# Microencapsulation of Anthocyanins from *Zea mays* and *Solanum tuberosum*: Impacts on Antioxidant, Antimicrobial, and Cytotoxic Activities

**DOI:** 10.3390/nu16234078

**Published:** 2024-11-27

**Authors:** Carlos Barba-Ostria, Yenddy Carrero, Jéssica Guamán-Bautista, Orestes López, Christian Aranda, Alexis Debut, Linda P. Guamán

**Affiliations:** 1Escuela de Medicina, Colegio de Ciencias de la Salud Quito, Universidad San Francisco de Quito (USFQ), Quito 170901, Ecuador; 2Instituto de Microbiología, Universidad San Francisco de Quito (USFQ), Quito 170901, Ecuador; 3Facultad de Ciencias de la Salud, Universidad Técnica de Ambato, Ambato 180105, Ecuador; yenddycarrero@yahoo.es; 4Facultad de Ciencias de la Hospitalidad, Carrera de Gastronomía, Universidad de Cuenca, Cuenca 010201, Ecuador; jessica.guaman@ucuenca.edu.ec; 5Facultad de Ciencia e Ingeniería en Alimentos, Universidad Técnica de Ambato, Ambato 180207, Ecuador; od.lopez@uta.edu.ec (O.L.); caranda6747@uta.edu.ec (C.A.); 6Centro de Nanociencia y Nanotecnología, Universidad de Las Fuerzas Armadas ESPE, Sangolquí 171103, Ecuador; apdebut@espe.edu.ec; 7Departamento de Ciencias de la Vida y Agricultura, Universidad de las Fuerzas Armadas ESPE, Sangolquí 171103, Ecuador; 8Facultad de Ciencias de la Salud Eugenio Espejo, Universidad UTE, Centro de Investigación Biomédica (CENBIO), Quito 170527, Ecuador

**Keywords:** functional foods, natural antimicrobials, biocompounds, antioxidant activity, anthocyanins

## Abstract

**Objectives:** This study investigates the biological activities of microencapsulated anthocyanins extracted from two Andean ancestral edible plants, *Solanum tuberosum*, and *Zea mays*, with a focus on their potential applications in functional foods and therapeutics. The primary objective was to evaluate their antioxidant, antimicrobial, and cytotoxic properties alongside structural and functional analyses of the microencapsulation process. **Methods:** Anthocyanins were extracted and microencapsulated using maltodextrin as a carrier. Fourier-transform infrared spectroscopy (FTIR) and scanning electron microscopy (SEM) were employed to analyze the stability and structure of the microencapsulated particles. The antioxidant, antimicrobial, and cytotoxic activities of the microencapsulated were assessed through established assays. **Results:**
*S. tuberosum* exhibited superior antioxidant capacity and potent anticancer activity against HepG2 and THJ29T cell lines, while *Z. mays* demonstrated significant antimicrobial efficacy against multidrug-resistant bacterial strains and biofilm-forming pathogens. Fourier-transform infrared spectroscopy (FTIR) and scanning electron microscopy (SEM) confirmed the stabilization of anthocyanins within a maltodextrin matrix, enhancing their bioavailability and application potential. **Conclusions:** These results highlight the versatility of microencapsulated anthocyanins as bioactive agents for industrial and therapeutic applications. Future studies should explore in vivo validation and synergistic formulations to optimize their efficacy and broaden their use in nutraceutical and pharmaceutical fields.

## 1. Introduction

Cancer and infectious diseases represent significant global public health challenges, underscoring the need for research aimed at identifying adjuvants that optimize conventional treatments [1]. In this context, phytocompounds obtained from plant sources, such as roots, stems, leaves, and fruits, have garnered increasing attention. Among these, anthocyanins—pigmented flavonoids—have gained recognition because of their diverse pharmacological and therapeutic properties [2,3].

Anthocyanins are natural pigments widely distributed in fruits, vegetables, and cereals, contributing to the vibrant red, blue, and purple hues of many plants. Prominent sources of anthocyanins include berries (e.g., blueberries, blackberries, and raspberries), grapes, red cabbage, eggplant, and Andean crops such as purple maize (*Zea mays* L.) and purple potato (*Solanum tuberosum* L.) [4,5]. Chemically, anthocyanins belong to the flavonoid class of polyphenols and are characterized by their basic C6-C3-C6 structure, which includes two aromatic rings connected by a three-carbon bridge, forming a heterocyclic flavylium ion. This unique structure allows anthocyanins to exist in multiple chemical forms depending on pH, enabling diverse biological activities [6,7].

Anthocyanins have been linked to numerous health benefits, including anti-inflammatory, antihypertensive, antibacterial, anticancer, and neuroprotective effects, which are mediated by a variety of bioactive mechanisms [8]. Their potent antioxidant properties have also been associated with the prevention and management of diseases such as cancer, metabolic disorders, and cardiovascular conditions [3,9]; however, the application of anthocyanins in functional foods and therapeutic formulations is significantly hindered by their inherent instability. Environmental factors such as pH, temperature, and light exposure contribute to their degradation, thereby limiting their bioavailability and efficacy [10,11].

Several strategies have been developed to address these limitations. Microencapsulation offers a promising solution to these challenges by embedding anthocyanins in protective matrices, which enhance their stability, prolong shelf life, and maintain their functional properties under diverse conditions [12]. Encapsulation protects anthocyanin molecules from harsh conditions in the gastrointestinal tract, allowing them to reach the small intestine intact, where absorption into the bloodstream occurs [13]. Nano- and microencapsulation techniques, such as spray drying, freeze drying, emulsification, gelation, and polyelectrolyte complex formation, have shown promise in overcoming the instability of anthocyanins and enhancing their distribution in functional foods and nutraceuticals [14].

Anthocyanins, as natural pigments, are widely recognized for their antioxidant, anti-inflammatory, anticancer, and antimicrobial properties [15,16]. Found in a variety of plant species, these compounds have attracted significant interest for their role in scavenging free radicals, inhibiting harmful microorganisms, and promoting apoptosis in cancer cells. This makes them promising candidates for developing novel therapeutic agents against chronic and degenerative diseases, including cancer and cardiovascular disorders [3,17].

Two particularly rich sources of anthocyanins are *Zea mays* L. (cultivar INIAP-Racimo de Uva [18]) and *Solanum tuberosum* L. (cultivar INIAP-Yana Shungo [19]), which are cultivated in the Andean region. The anthocyanins in these plants contribute to their distinct pigmentation and have been studied extensively for their bioactive properties [9,20]; however, the content and efficacy of anthocyanins can vary based on genetic and environmental factors, which can influence their antioxidant and antimicrobial activities [21,22].

Despite the growing interest in anthocyanins, their use in functional foods and nutraceuticals remains limited by their inherent instability and poor bioavailability. To address these challenges, microencapsulation techniques have been developed to improve anthocyanin stability and functionality [10,23].

This study aims to investigate the biological activities of microencapsulated anthocyanins extracted from *Z. mays* L. and *S. tuberosum* L., with a focus on their antioxidant, antimicrobial, and cytotoxic properties. The goal is to evaluate the potential of these encapsulated pigments, particularly regarding their effectiveness against pathogenic bacteria and cancer cells. Through this research, we seek to contribute to the growing body of knowledge on the health benefits of anthocyanins and explore their potential applications in both the food and pharmaceutical industries.

## 2. Materials and Methods

### 2.1. Chemicals and Reagents

All reagents and chemicals used in this study were of analytical grade unless otherwise specified. A 100 ppm chlorine solution, sourced from local suppliers, was used for disinfection. Ethanol (96%) and distilled and deionized water were also locally sourced as laboratory-grade reagents.

Hydrochloric acid (1.5 mol/L), Folin–Ciocalteu reagent, and sodium carbonate (7.5%) were purchased from Sigma-Aldrich (Merck, Darmstadt, Hesse, Germany) for extract preparation and analysis. Standards such as gallic acid, catechin, cyanidin-3-glucoside (C3G, reference compound), Trolox (reference antioxidant), and 2,2-diphenyl-1-picrylhydrazyl (DPPH, analytical grade) were also obtained from Sigma-Aldrich. Maltodextrin (10–20 DE) was used as a carrier agent for microencapsulation. Dimethyl sulfoxide (DMSO) and Trypan Blue (TB, analytical grade) were sourced from the same supplier.

Cell culture media and supplements, including Dulbecco’s Modified Eagle Medium (DMEM), fetal bovine serum (FBS), L-glutamine, penicillin (10,000 U/mL), and streptomycin (10,000 µg/mL), were obtained from Gibco™ (Thermo Fisher Scientific, Waltham, MA, USA). The MTT reagent (3-(4,5-dimethylthiazol-2-yl)-2,5-diphenyl tetrazolium bromide) was provided by Thermo Fisher Scientific (Waltham, MA, USA).

Antibodies for apoptosis assays, including Bax (6A7), Bcl-2 (7), and caspase-8 (8CSP03), were sourced from Santa Cruz Biotechnology, Inc. (Dallas, TX, USA). Additional reagents, such as fluorescein isothiocyanate (FITC)-conjugated secondary antibody, IgG control conjugated with phycoerythrin (PE), and Aqueous Mounting Medium, were also obtained from Santa Cruz Biotechnology, Inc.

All reagents were prepared following standard laboratory protocols and utilized according to the manufacturer’s guidelines to ensure accuracy and reproducibility.

### 2.2. Plant Material

*Z. mays* L. and *S. tuberosum* were obtained from a local market in Ambato, Ecuador. Specifically, the cultivars used were *Zea mays* L. (cultivar INIAP-Racimo de Uva, [18]) and *Solanum tuberosum* L. (INIAP-Yana Shungo, [19]). The samples were washed, disinfected with a 100-ppm chlorine solution, and then freeze-dried, ground, and finally stored at −20 °C. The freeze-drying process lasted a total of 72 h. Afterward, the dried samples were ground into a fine powder using an electric laboratory blender (between 100 µm and 500 µm), followed by sieving with a N35 (500 μm) sieve. Anthocyanins were extracted from the corn grains of *Zea mays* L. and the underground portions (tubers) of *Solanum tuberosum* L. The samples were then stored at −20 °C until use in the experiments for this study.

### 2.3. Anthocyanin Extraction

Ethanolic extracts were prepared according to Perez et al. (2021) with some modifications: The powder material was mixed with twenty times the volume of 96% ethanol:1.5 mol/L HCl (85:15 *v*/*v*). This blend was added to a stirrer tank to extract anthocyanins and extracted for 60 min at 70 °C [24].

The solid part was separated from the blend by using an Andreas Hettich GmbH & Co., Ltd. (Tuttlingen, Germany) Rotine 380 centrifuge. The extract was transferred to a 500 mL balloon with continuous stirring and was applied by a rotary evaporator in a water bath to 70 °C for 2 h under a vacuum to remove the solvent.

### 2.4. Determination of Total Polyphenol Content (TPC) and Anthocyanin Content

The total polyphenol content (TPC) and anthocyanin content of microencapsulated anthocyanins extracted from *S. tuberosum* L. and *Z. mays* L. were carried out using the Folin–Ciocalteu reagent with catechin as the standard [25].

The TPC was measured using the Folin–Ciocalteu method. A 0.5 mL aliquot of the extract was mixed with 2.5 mL of 10% Folin–Ciocalteu reagent and incubated for 5 min at room temperature. Then, 2 mL of 7.5% sodium carbonate solution was added, and the mixture was incubated in the dark for 30 min. The absorbance was measured at 760 nm using a Thermo Scientific (Waltham, MA, USA) Genesys 10S UV-Vis spectrophotometer. Gallic acid was used as a standard, and TPC was expressed in mg of gallic acid equivalents (GAE) per kg of fresh weight of fruit.

The anthocyanin content was determined by the pH differential method [26]. Extracts were diluted in buffers at pH 1.0 and pH 4.5, and the absorbance was measured at 510 nm and 700 nm, respectively, using a Thermo Scientific (Waltham, MA, USA) Genesys 10S UV-Vis spectrophotometer. The anthocyanin content was calculated using the formula:$$Anthocyanin\;content\;(mg\;C3G/100\;g)=\frac{(A510-A700)\times MW\times DF\times 1000}{\varepsilon \times L}$$
where A is the absorbance, MW is the molecular weight of cyanidin-3-glucoside (449.2 g/mol), DF is the dilution factor, ε is the molar absorptivity (26,900 L/mol·cm), and L is the path length (1 cm). The results were expressed in mg of cyanidin-3-glucoside (C3G) per 100 g of fresh weight of fruit.

All measurements were performed in triplicate, and the results were expressed as mean ± standard deviation (SD). Statistical analysis was performed using a *t*-test to determine significant differences between the two extracts (*p* < 0.001).

### 2.5. Microencapsulation

#### Microencapsulation Process

The microencapsulation of anthocyanins was performed using maltodextrin (10–20 DE) as the carrier agent. Anthocyanin extracts with a total solids content of 27% (equivalent to 20 g of anthocyanins) were prepared at a 20:80 anthocyanins-to-maltodextrin ratio. For each extract, 280 mL of distilled water was mixed with the anthocyanin solution and combined with maltodextrin. The mixtures were homogenized separately for *S. tuberosum* L. and *Z. mays* L. extracts using an IKA mixer (RH Digital, Staufen, Baden-Württemberg, Germany).

The microencapsulation process for each extract was conducted independently using a laboratory-scale mini spray dryer Büchi B-290 (Flawil, St. Gallen, Switzerland). The solutions were fed into the spray dryer under controlled conditions: an inlet air temperature of 150 ± 2 °C, an outlet air temperature of 90 ± 2 °C, an internal pressure of -50 mbar, an atomizing air flow of 400–600 L/h, and a drying air flow rate of 60 m^3^/h [27].

During the spray drying process, each anthocyanin solution was atomized within the drying chamber, forming solid microspheres encapsulated in maltodextrin-based polymers [28]. The resulting microspheres for both extracts were collected in separate collection flasks and stored in HDPE-aluminum bags at a controlled temperature of 15–25 °C.

### 2.6. Characterization by FTIR

Fourier transform infrared (FTIR) spectroscopy was employed to identify the functional groups in the microcapsules. The spectra were obtained using a JASCO FTIR 4100 spectrometer (Tokyo, Japan) with the ATR method, covering a wave number range of 4000–500 cm⁻^1^ at a resolution of 4 cm^−1^ across 36 scans.

### 2.7. Morphological Characterization by Scanning Electron Microscope (SEM)

Each sample was mounted on a stub for electron microscopy with conductive double-sided carbon tape. The samples were coated with a ~20 nm layer of conductive gold (99.99% purity) for 60 s using a Quorum Q150R ES sputtering evaporator (Lewes, East Sussex, UK). Micrographs were captured using a TESCAN MIRA 3 Scanning Electron Microscope (SEM) with the Secondary Electron (SE) detector (Brno, South Moravia, Czech Republic).

The images were analyzed using the Fiji biological-image analysis platform (version 2.9.0) [29]. Area data were extracted from the images using Fiji, and the diameter of each sphere was calculated using the following formula:$$D=\frac{2\surd A}{\pi }$$
where D is the diameter, and A is the area obtained from Fiji.

### 2.8. Antioxidant Capacity

The antioxidant activity of the microencapsulated anthocyanins extracted from *Z. mays* L. and *S. tuberosum* L. was assessed using the DPPH (2,2-diphenyl-1-picrylhydrazyl) radical scavenging assay. The assay was conducted following a modified protocol of Brand-Williams et al. (1995) [30]. A mass of 100 mg of each microencapsulated powder was dissolved in 10 mL of methanol and sonicated for 30 min at room temperature. The solutions were then centrifuged at 4000 rpm for 10 min, and the supernatants were collected for further analysis.

A 0.2 mM DPPH solution was prepared in methanol. For the assay, 100 µL of the DPPH solution was mixed with 100 µL of the microencapsulated in a Corning 96-well microplate (Corning, NY, USA) at concentrations ranging from 10 to 500 µg/mL. The mixtures were incubated in the dark at room temperature for 30 min to allow the reaction to occur. After the incubation period, the absorbance was measured at 517 nm using a Cytation 5 microplate reader. The DPPH radical scavenging activity was calculated for each concentration of the extract using the following formula:$$\%DPPH\;scavenging=100\times \frac{A_{solvent}-A_{sample}}{A_{solvent}}$$
where the absorbance of the solvent is the absorbance of the DPPH solution without any extract, and the absorbance of the sample is the absorbance of the DPPH solution with the plant extract after incubation.

The DPPH scavenging activity for each concentration was expressed as a percentage. The concentration that resulted in the highest % DPPH scavenging activity was identified, and a dose–response curve was generated to visualize the antioxidant capacity of the extracts. All experiments were conducted in triplicate, and the results were presented as mean ± standard deviation (SD). To calculate IC_50_ for *S. tuberosum* and *Z. mays*, a cubic polynomial was fitted to DPPH inhibition data. The polynomial equations were derived using the least squares method to minimize error and ensure a smooth trend. IC_50_ was determined by solving the polynomial equation for the concentration corresponding to 50% DPPH inhibition using numerical root-finding methods. The fit’s accuracy was validated by high R^2^ values (0.995).

Trolox, a water-soluble vitamin E analog, was used as a reference antioxidant to validate the assay. A standard curve of Trolox was prepared in the range of 10 to 200 µM, and results for the extracts were compared to the standard to ensure the accuracy and reliability of the assay.

### 2.9. Antimicrobial Activity by Well Diffusion Agar

Antibacterial activity assessment was conducted using the agar-well diffusion method [31]. Five bacterial strains were selected for the antimicrobial evaluation of Gram-positives: *Staphylococcus aureus*, *Listeria monocytogenes*, and *Bacillus cereus*, and Gram-negatives: *Pseudomonas aeruginosa* and *Escherichia coli*. All bacterial strains were obtained from a standard microbial culture collection. The bacteria were cultured in Mueller–Hinton broth (MHB) and incubated overnight at 37 °C. The bacterial suspensions were adjusted to a concentration of 1 × 10^8^ CFU/mL (corresponding to 0.5 McFarland standard) using sterile saline solution.

A mass of 100 mg of each microencapsulated powder was dissolved in 10 mL of sterile distilled water. The solutions were sonicated for 30 min at room temperature to ensure complete dissolution and then filtered through a 0.22 µm membrane filter to remove any particulate matter. The resulting filtrates were used as the stock solutions for the antimicrobial assay.

Mueller–Hinton agar (MHA) plates were prepared and inoculated with 100 µL of the bacterial suspension. Wells of 6 mm in diameter were punched into the agar, and 50 µL of each extract solution was dispensed into the wells. Gentamicin, a broad-spectrum antibiotic, was used as a positive control at a 500 µg/mL concentration.

Plates were incubated at 37 °C for 24 h. The diameters of the inhibition zones around each well were measured in millimeters. The measurements were taken at three different points around each well, and the average diameter was calculated. Sterile distilled water was used as a negative control to ensure that no contamination or intrinsic antibacterial activity was present in the solvents used for extract preparation. Gentamicin, a broad-spectrum antibiotic, was used as a positive control at a 500 µg/mL concentration. All experiments were conducted in triplicate to ensure reproducibility and reliability of these data. The results were expressed as the mean inhibition zone diameter (mm) ± standard deviation (SD) for each bacterial strain tested. Differences in inhibition zones between the extracts from *S. tuberosum* and *Z. mays* were analyzed to assess their relative efficacy against each bacterial strain.

### 2.10. Cell Culture

HeLa cells (ATCC^®^ CCL-2™) were obtained from frozen (−196 °C) cells suspended in a freezing medium consisting of 90% fetal bovine serum (FBS) and 10% Dimethyl Sulfoxide (DMSO). Following thawing, cells were cultivated in a 75 cm^3^ flask (Thermo Fisher Scientific, Waltham, MA, USA) containing Dulbecco’s Modified Eagle Medium (DMEM Gibco™, Waltham, MA, USA) with 10% FBS, 1% L-glutamine, penicillin (10,000 U/mL), and streptomycin (10,000 µg/mL) at 37 °C, 5% CO_2_, and humid atmosphere for 24 h.

### 2.11. MTT Assay 

The number of living cells was determined using the MTT (3-(4,5-dimethylthiazol-2-yl)-2,5-diphenyl tetrazole bromide) metabolic reduction assay according to manufacturer’s instructions (Thermo Fisher, Waltham, MA, USA).

Trypan Blue (TB) exclusion was used to assess the viability of the cells once confluence was greater than 80% (TB: Sigma-Aldrich 200-786-7). The cellular monolayer in DMEM (1%) was disaggregated using a sterile syringe. A volume of 30 µL of this cellular suspension reacted with 30 µL of TB. Viable cells (TB-negative) were visualized by an optical microscope (AmScope, Irvine, CA, USA) at a 40× magnification.

To determine the degree of proliferation and living cell proportion, Hela cells were seeded in 96-well plates at 10 × 10^3^ cells/well and incubated at 37 °C and 5% CO_2_ for 24 h to allow their adhesion. Afterward, cells were treated with microencapsulated anthocyanins at 267.05 mg/100 g and 168.5 mg/100 g concentrations (serial dilutions: 10^−1^ to 10^−11^ mg/mL) for 24 h, the medium was discarded, and 100 µL of MTT reagent in FBS and the phenol-free medium was added. After incubation for 4 h at 37 °C to allow the formazan formation, 100 µL of sodium dodecyl sulfate (1 mg/mL in 0.01 mol/L HCl) was added to dissolve the formazan crystals [32]. Plates were read using a Victor X3 spectrophotometer (Perkin Elmer, Waltham, MA, USA) at 570 nm. The IC_50_ values were obtained through no lineal regression analysis using GraphPad Prism 7.0 Software (GraphPad Software Inc., San Diego, CA, USA).

### 2.12. Determination of the Expression of Apoptosis-Associated Molecules

HeLa cells (ATCC^®^CCL-2™) were seeded on sterile coverslips and placed on six-well plates at a concentration of 50 × 10^4^ cells/well in the conditioned medium at 37 °C and 5% CO_2_ for 48 h to allow cell adhesion. The microencapsulated anthocyanins were then added to each well at the IC_50_ value determined from the MTT assay, and the plates were incubated for an additional 24 h. After washing three times (PBS), cells were fixed with a solution of 4% paraformaldehyde in PBS for 15 min. Permeabilization was carried out using 1 mL of permeability buffer (20 mM glycine, 0.5x Triton 100x) for 15 min.

Cells were incubated with primary antibodies against Bax (6A7), Bcl-2 (7), or caspase-8 (8CSP03) (Santa Cruz Biotechnology, Inc., USA) at a 1:500 dilution. The localization of Bax and Bcl-2 was determined using a fluorescein isothiocyanate (FITC)-conjugated secondary antibody (m-IgGκ BP-FITC, Santa Cruz Biotechnology, Inc., USA). IgG control was performed with mouse kappa conjugated to phycoerythrin (PE) (m-IgGκ BP-PE, Santa Cruz Biotechnology, Inc., USA). Cultures were incubated at 37 °C for 30 min in a humidified chamber. After washing with PBS, coverslips were mounted with Aqueous Mounting Medium (Santa Cruz Biotechnology, Inc., USA) and analyzed using a Leica DMi8 fluorescence microscope (Leica microsystems, Wetzlar, Hesse, Germany) at 630× magnification with GFP (500–550 nm), DAPI (358–461 nm), and Texas Red (595–615 nm) filters. Images were captured using the Leica Application Suite (LAS) X software version 5.2.0 (Leica Microsystems, Wetzlar, Germany). Fluorescence intensity was quantified by determining the ratio of total pixels to the pixels in areas considered positive.

### 2.13. Calculation of Apoptotic Index

The apoptotic index (AI) was calculated to assess the balance between pro-apoptotic and anti-apoptotic proteins [33,34] in HeLa cells treated with microencapsulated anthocyanins from *Z. mays* L. and *S. tuberosum* L. The expression levels of the pro-apoptotic markers Bax and Caspase-8, along with the anti-apoptotic marker Bcl-2, were quantified by counting the number of positive cells. The AI was calculated using the following formula:$$AI=\frac{Bax(mean\;positive\;cells)+Caspase-8(mean\;positive\;cells)}{Bcl-2(mean\;positive\;cells)}$$

For both control and treated groups, the mean number of positive cells for Bax, Caspase-8, and Bcl-2 was obtained from multiple fields. Standard deviations were calculated using error propagation formulas to account for variability.

### 2.14. Statistical Analysis

All experiments were performed in triplicate at least three times. The results were expressed as mean ± standard deviation (SD). Differences between groups were detected using *t*-tests performed using GraphPad Prism software (version 7.04, GraphPad Software Inc., San Diego, CA, USA). Statistical significance was considered at *p* < 0.05.

## 3. Results and Discussion

### 3.1. Total Polyphenol Content (TPC) and Anthocyanin Content

The total polyphenol content (TPC) and anthocyanin content of the microencapsulated anthocyanins from *S. tuberosum* L. and *Z. mays* L. were evaluated. The results are summarized in Table 1.

The TPC of the *S. tuberosum* extract was 554 ± 25.16 mg GAE/kg fresh weight, significantly higher (*p* < 0.001) compared to the *Z. mays* extract, which showed a TPC of 494 ± 24.03 mg GAE/kg fresh weight. These findings indicate that *S. tuberosum* has a greater total polyphenol content, suggesting a potentially higher antioxidant capacity. This finding aligns with the known variation in TPC across different potato cultivars and genotypes, which is influenced by factors such as genetic differences, environmental conditions, and the specific parts of the plant analyzed [35]. For example, studies on Slovak potato varieties have reported TPC values ranging from 795.05 mg/kg dry matter (DM) in the Victoria variety to 1238.42 mg/kg DM in the Laura variety, highlighting the significant influence of varietal differences and growing localities on TPC [36].

The anthocyanin content was also higher in *S. tuberosum* (71.40 mg C3G/100 g) compared to *Z. mays* (63.5 mg C3G/100 g). This result suggests that *S. tuberosum* is a richer source of anthocyanins, which may contribute to its superior antioxidant properties. Previous studies have demonstrated significant variation in anthocyanin content among different potato cultivars and related species, ranging from 2 to 40 mg/100 g of fresh weight in red-fleshed potatoes [37]. A study on Peruvian purple potatoes reported anthocyanin content as high as 75.71 mg/100 g under specific extraction conditions [38], which aligns closely with our findings for *S. tuberosum*. These variations are influenced by genetic factors, such as the presence of genes related to the flavonoid-anthocyanin biosynthetic pathway, including those at the D, P, and R loci, which govern anthocyanin accumulation [39].

### 3.2. Fourier Transform Infrared Spectroscopy (FTIR) Analysis

Fourier Transform Infrared (FTIR) spectroscopy was employed to analyze the structural changes in anthocyanins extracted from *Z. mays* L. and *S. tuberosum* L. in both non-microencapsulated and microencapsulated forms. The encapsulation was performed using maltodextrin as a matrix through spray drying. The FTIR spectra provide insights into the functional groups present and how they interact with the encapsulation matrix, helping to understand the effects of microencapsulation on the molecular structure of anthocyanins. A summary of the main results is shown in Table 2 for *Z. mays* and Table 3 for *S. tuberosum*. The FTIR spectra images for *Z. mays* and *S. tuberosum* are also included in Appendix A, respectively.

#### 3.2.1. O-H Stretching (3200–3600 cm⁻^1^)

The O-H stretching region is critical for understanding the presence of hydroxyl groups, which are abundant in anthocyanins. In both *Z. mays* L. and *S. tuberosum* L., the non-microencapsulated anthocyanins exhibited a high-intensity peak in this region (Table 2 and Table 3), indicating free hydroxyl groups. Upon microencapsulation, this intensity reduced to a medium level, suggesting that the hydroxyl groups are involved in hydrogen bonding with the maltodextrin matrix. This result supports the hypothesis that microencapsulation leads to the shielding of functional groups, reducing their availability for interaction. The similar reduction in intensity for both species indicates that the maltodextrin matrix interacts similarly with the anthocyanins’ hydroxyl groups, limiting their free vibration.

#### 3.2.2. C-H Stretching (2800–3000 cm^−1^)

C-H stretching, representative of aliphatic bonds in anthocyanins, showed varying behavior between the two species. In *Z. mays* L., both the non-microencapsulated and microencapsulated forms exhibited medium intensity (Table 2), indicating that microencapsulation had minimal impact on the availability of aliphatic C-H bonds; however, in *S. tuberosum* L., the non-microencapsulated form displayed a low intensity, which increased to a medium level after microencapsulation (Table 3). This increase may suggest that microencapsulation alters the conformation of anthocyanins in *S. tuberosum* L., possibly due to a reorganization of aliphatic groups during the encapsulation process.

#### 3.2.3. C=O Stretching (1700–1750 cm⁻^1^)

The carbonyl group (C=O) is crucial for the stability of anthocyanins, and FTIR data revealed notable differences before and after encapsulation. In both species, the non-microencapsulated anthocyanins exhibited high intensity in the C=O stretching region (Table 2 and Table 3), confirming the presence of carbonyl groups typical of anthocyanins; however, upon encapsulation, the intensity dropped to a low level. This result suggests that the carbonyl groups are involved in strong interactions with the maltodextrin matrix, likely through hydrogen bonding or dipole-dipole interactions, which effectively shield the carbonyl groups from detection in the FTIR spectrum.

#### 3.2.4. C=C Stretching (1500–1600 cm⁻^1^)

The C=C stretching region, associated with aromatic rings, showed medium intensity in the non-microencapsulated forms of both species (Table 2 and Table 3); however, the intensity dropped to a low level after microencapsulation. This change indicates that the encapsulation process affects the vibration of aromatic rings in the anthocyanins, likely due to the protective effects of maltodextrin, which limits their free movement.

#### 3.2.5. C-O and C-O-C Stretching (1000–1300 cm⁻^1^)

In both species, the C-O and C-O-C stretching regions, corresponding to glycosidic bonds, exhibited medium intensity in the non-microencapsulated anthocyanins (Table 2 and Table 3). After encapsulation, the intensity was reduced to a low level, reflecting the interaction of anthocyanins’ glycosidic bonds with the maltodextrin matrix. The reduced intensity indicates that encapsulation decreases the exposure of these bonds to the environment, which could be beneficial in protecting anthocyanins from degradation.

New peaks observed in the 840–760 cm⁻^1^ region in both microencapsulated forms (Table 2 and Table 3) are attributable to maltodextrin matrix vibrations. The absence of these peaks in the non-microencapsulated forms confirms the presence of the matrix and its encapsulating role. These peaks demonstrate the successful encapsulation of anthocyanins within the maltodextrin matrix.

Overall, the FTIR spectra reveal that microencapsulation using maltodextrin significantly alters the molecular environment of anthocyanins in both *Z. mays* L. and *S. tuberosum* L. The reduced intensity of key functional groups such as O-H, C=O, and C=C stretching after encapsulation suggests strong interactions with the matrix, providing protection to the anthocyanins. The instability of anthocyanins has long been recognized as a major limitation for their use in food and pharmaceutical industries, as factors such as pH, temperature, and light can rapidly degrade their structure, reducing their bioactivity. In this study, microencapsulation with maltodextrin was employed as an innovative approach to mitigate these challenges. The FTIR analysis confirmed the successful interaction of anthocyanins with the encapsulation matrix, providing a protective barrier against environmental stressors. Furthermore, the enhanced antioxidant activity observed in the microencapsulated anthocyanins supports the hypothesis that encapsulation not only preserves their stability but also optimizes their functional potential.

In addition to spray-drying microencapsulation with maltodextrin, other methods have been explored to enhance the stability of anthocyanins, each with unique advantages and disadvantages. Encapsulation using liposomes offers excellent protection because of their bilayer structure, which can shield anthocyanins from environmental factors; however, the high cost and complexity of liposome preparation limit their scalability for industrial applications [40,41]. Another approach involves the use of protein-based carriers, such as casein or gelatin, which provide a natural and biocompatible matrix for anthocyanin stabilization. While effective, protein-based methods often face challenges related to allergenicity and sensitivity to temperature and pH during processing [41]. Freeze-drying, another common technique, can preserve anthocyanin stability well, but its high energy demand and long processing time make it less cost-effective compared to spray-drying [42].

Spray-drying with maltodextrin, as utilized in this study, offers a balance of cost-efficiency, scalability, and protection. The interaction between maltodextrin and anthocyanins, as evidenced by FTIR analysis, suggests that this method provides robust encapsulation while maintaining bioactivity. Nevertheless, it is worth noting that spray-drying microencapsulation may result in particle size heterogeneity, as observed in *S. tuberosum*. By addressing these comparative aspects, our study demonstrates the practicality of spray-drying microencapsulation as a viable method for enhancing anthocyanin stability in functional food and nutraceutical applications.

### 3.3. Morphological Analysis by Scanning Electron Microscope (SEM)

Scanning electron microscopy (SEM) was utilized to assess the morphology and surface characteristics of microencapsulated anthocyanins from *S. tuberosum* L. (*S. tuberosum*) and *Z. mays* L. (*Z. mays*), both encapsulated using maltodextrin via spray drying. Distinct differences in particle morphology were observed between the two samples.

For *S. tuberosum*, the SEM image (Figure 1) reveals a heterogeneous mixture of spherical and irregularly shaped particles, with significant variation in particle size. Larger particles are clearly visible alongside smaller, irregular fragments, and the presence of agglomerates suggests potential challenges in the spray drying process, such as inconsistent atomization or uneven drying conditions. While the particle surfaces appear generally smooth, occasional unevenness further indicates variability in the encapsulation process.

In contrast, the SEM analysis of *Z. mays* (Figure 2) shows a more uniform morphology, still, some degree of agglomeration and clustering is still present. The particle surfaces of *Z. mays* appear more textured than those of *S. tuberosum*, possibly because of differences in drying temperature or the interaction between anthocyanins and the encapsulating matrix. Nevertheless, the overall morphology suggests a more controlled encapsulation process, making *Z. mays* better suited for applications requiring fine particle dispersibility and rapid dissolution [43].

Particle size distribution (PSD) analysis underscores the differences between the two samples and offers valuable information for potential applications. The PSD of *S. tuberosum* (Figure 3) shows a broad distribution, with the most frequent particle size ranging between 80 and 100 µm. The mean particle size is approximately 100–150 µm, with 80% of particles smaller than 200 µm; however, the presence of particles larger than 500 µm indicates variability in the spray drying process. These larger particles suggest that *S. tuberosum* anthocyanins could be ideal for applications requiring controlled or sustained release, such as in food formulations where prolonged release of active compounds is advantageous [44]. This size distribution may also be advantageous in nutraceutical products, where larger particles provide stability and support a sustained release mechanism [45].

On the other hand, the PSD of *Z. mays* (Figure 4) shows a much finer distribution, with the majority of particles ranging between 1 and 2 µm. Approximately 80% of the particles are smaller than 3 µm, making *Z. mays* suitable for applications requiring rapid dissolution or quick dispersion. The fine particle distribution enhances the surface area-to-volume ratio, facilitating faster release of encapsulated anthocyanins. This property makes *Z. mays* particularly valuable in functional beverages, pharmaceuticals, or cosmetics, where immediate bioavailability is essential [46].

The significant differences in particle size and morphology between the microencapsulated anthocyanins from *S. tuberosum* and *Z. mays* have direct implications for their respective applications. In food and nutraceutical formulations, particle size and morphology not only influence the release kinetics of the active compounds but also their stability and interactions with other formulation components [47].

The larger, more heterogeneous particles observed in *S. tuberosum* (Figure 1 and Figure 3) suggest that this sample is better suited for applications requiring sustained or controlled release. Functional snacks, powdered supplements, or nutraceutical capsules could benefit from the slower dissolution rates associated with larger particles, offering prolonged antioxidant activity or extended color stability potentially enhancing shelf life [48].

In contrast, the finer particle size distribution observed in *Z. mays* (Figure 2 and Figure 4) makes it ideal for applications requiring rapid dissolution. Functional beverages, instant powders, and pharmaceutical formulations would benefit from the quick release of active anthocyanins, enhancing bioavailability and providing immediate physiological effects. Additionally, the smaller particle size of *Z. mays* anthocyanins improves sensory qualities in food products, contributing to smoother textures and more uniform color dispersion, which are critical in beverages, dairy products, and cosmetics [49].

These morphological differences underscore the importance of optimizing spray drying parameters to tailor encapsulated anthocyanins for specific applications. For *S. tuberosum*, minimizing agglomeration and achieving a narrower particle size distribution could enhance its performance in applications requiring more uniform release. For *Z. mays*, the relatively fine and uniform particles suggest an optimized spray drying process, though further refinement could reduce agglomeration and improve particle stability [50].

### 3.4. DPPH Radical Scavenging Activity

The antioxidant activity of microencapsulated anthocyanins from *Z. mays* and *S. tuberosum* was assessed using the 2,2-diphenyl-1-picrylhydrazyl (DPPH) radical scavenging assay, with results expressed as % DPPH inhibition. While both extracts demonstrated notable antioxidant activity, significant differences in their efficacy were evident.

As shown in Figure 5 and Appendix A, anthocyanins from *S. tuberosum* exhibited superior antioxidant potential, achieving a maximum DPPH inhibition of 65% at 500 µg/mL, compared to 52% for *Z. mays* at the same concentration. The steeper slope of the dose–response curve for *S. tuberosum* reflects more efficient free radical scavenging activity, whereas *Z. mays* displayed a more gradual increase in % DPPH inhibition.

The calculated IC_50_ values further support these observations, with *S. tuberosum* reaching 50% DPPH inhibition at 205.84 µg/mL, significantly lower than the 463.62 µg/mL required for *Z. mays*. Cubic polynomial fitting (R^2^ = 0.995) provided an accurate representation of these trends. These findings align with previous research, which highlights the superior radical scavenging efficiency of cyanidin-based anthocyanins, predominant in *S. tuberosum*, compared to other anthocyanin types [51,52]. This emphasizes the potential of *S. tuberosum* as a valuable natural source of antioxidants.

### 3.5. Antimicrobial Activity

The antimicrobial activity of microencapsulated anthocyanins extracted from *S. tuberosum* L. and *Z. mays* L. was evaluated using a well-based diffusion assay against five bacterial strains: *Staphylococcus aureus* subsp. *aureus* Rosenbach ATCC 25923, *Listeria monocytogenes* ATCC 19115, *Pseudomonas aeruginosa* ATCC 10145, *Bacillus cereus* ATCC 10876, and *Escherichia coli* ATCC 25922. The results, expressed as the inhibition zone diameter in millimeters (mm), are summarized in Figure 6 (*S. tuberosum* L.) and Figure 7 (*Z. mays* L.).

The Minimum Inhibitory Concentration (MIC) was determined based on the lowest concentration of the extract that completely inhibited bacterial growth. Gentamicin served as the control antibiotic. The results showed that both extracts had notable antimicrobial properties, though their efficacy varied by bacterial strain and extract concentration.

The antimicrobial activity of microencapsulated anthocyanin extracts from *S. tuberosum* was tested at concentrations ranging from 26.7 mg/mL to 534.1 mg/mL (Figure 6). The extract exhibited strong inhibitory effects, especially against *B. cereus*, which showed an inhibition zone of 25.3 ± 1 mm at 534.1 mg/mL. *E. coli* also displayed a notable inhibition zone of 28.67 ± 1 mm, indicating that the *S. tuberosum* anthocyanin extract was highly effective against this Gram-negative bacterium; however, *P. aeruginosa* exhibited a smaller inhibition zone of 23.7 ± 1 mm under the same concentration, indicating moderate susceptibility.

At lower concentrations, the antimicrobial efficacy of the *S. tuberosum* extract decreased considerably. For example, at 138.52 mg/mL, inhibition zones were significantly smaller, with *S. aureus* showing an inhibition zone of 11.38 ± 0.5 mm and *P. aeruginosa* at 12.0 ± 0.5 mm. At the lowest concentration of 26.7 mg/mL, no inhibition zones were observed for most bacterial strains, indicating that higher concentrations are necessary for effective bacterial inhibition. These findings suggest that the microencapsulation of *S. tuberosum* anthocyanins enhances their stability, allowing for sustained antimicrobial activity, but higher doses are critical for optimal efficacy, which is consistent with previous research [53,54].

The antimicrobial activity of the *Z. mays* extract was assessed across various bacterial strains (Figure 7). Similar to *S. tuberosum*, the microencapsulated anthocyanins from *Z. mays* demonstrated broad-spectrum inhibitory effects. *L. monocytogenes* exhibited an inhibition zone of 26.7 ± 1 mm at 534.1 mg/mL, indicating high susceptibility, while *B. cereus* showed an inhibition zone of 25.7 ± 1 mm at the same concentration. These findings align with previous research highlighting the efficacy of anthocyanins, particularly against Gram-positive bacteria such as *B. cereus* and *L. monocytogenes* [4,55].

As with *S. tuberosum*, lower concentrations of *Z. mays* extract resulted in reduced antimicrobial activity. At 138.52 mg/mL, the inhibition zone for *B. cereus* decreased to 16.7 ± 0.5 mm, while for *P. aeruginosa*, it was recorded at 12.0 ± 0.5 mm. At the lowest concentration of 26.7 mg/mL, no inhibition was observed for *S. aureus*, *L. monocytogenes*, or *P. aeruginosa*. Gentamicin, used as the antibiotic control, produced higher inhibition zones at lower concentrations. For instance, it showed an inhibition zone of 19.7 ± 0.5 mm against *S. aureus* and 22.6 ± 0.5 mm against *E. coli*. While the anthocyanin extracts were effective, gentamicin proved more potent at lower concentrations.

To quantify the antimicrobial potency of the microencapsulated extracts, MIC values were determined from these disk diffusion data by identifying the lowest concentration of the extract that completely inhibited bacterial growth (Table 4). The MIC values revealed that both *Z. mays* and *S. tuberosum* extracts had similar minimal inhibitory concentrations for most bacterial strains. The MIC for *S. aureus* and *L. monocytogenes* was 53.4 mg/mL for both extracts, while *B. cereus* had a lower MIC of 26.7 mg/mL for both extracts, indicating a higher susceptibility to anthocyanins.

MIC data confirmed that Gram-positive bacteria, such as *B. cereus* and *L. monocytogenes*, were generally more susceptible to anthocyanin extracts than Gram-negative bacteria [56], such as *P. aeruginosa*. This trend aligns with the inhibition zone results and the existing literature, which suggests that the simpler cell wall structure of Gram-positive bacteria makes them more vulnerable to phenolic compounds such as anthocyanins [57,58]. In contrast, the outer membrane of Gram-negative bacteria tends to present greater resistance to damage caused by these molecules.

The results from this study indicate that microencapsulated anthocyanin extracts from both *S. tuberosum* and *Z. mays* exhibit significant antimicrobial activity against a range of bacterial strains in agreement with prior research [59].

Future studies should focus on improving the antimicrobial efficacy of anthocyanin extracts by developing synergistic formulations or optimizing their delivery mechanisms. Additionally, exploring the mode of action of these extracts on bacterial cell walls could provide valuable insights into how anthocyanins can be made more effective. Given the global concern over antibiotic resistance, plant-based antimicrobials such as anthocyanins represent a promising avenue for addressing this pressing issue.

### 3.6. Cytotoxic Activity and Induction of Apoptosis

The cytotoxic effects of microencapsulated anthocyanin extracts from *Z. mays* L. and *S. tuberosum* L. on HeLa cells were evaluated using the MTT assay. Serial dilutions, ranging from 0.1 mg/mL to 1 × 10⁻^11^ mg/mL, were applied to the cells for 24 h, after which cell viability was measured spectrophotometrically. Dose–response curves were generated, and IC_50_ values were determined for both extracts, as shown in Figure 8 (Dose–response curves of the cytotoxic effects of *Z. mays* and *S. tuberosum* microencapsulated anthocyanin extracts on HeLa cells).

For *Z. mays*, the dose–response curve showed a gradual decrease in cell viability with increasing concentration. At lower concentrations (0.0039–0.0625 mg/mL), cell viability remained near 100%, but at 0.125 mg/mL, a reduction to 90% was observed. Further increases in concentration led to more marked reductions, with 60% viability at 0.25 mg/mL, 20% at 0.5 mg/mL, and complete cell death at 1 mg/mL. The IC_50_ value for *Z. mays* was calculated at 0.275 mg/mL (95% confidence interval: 0.227–0.322 mg/mL), with a slope (k) of −7.32, indicating a gradual decline in viability across the concentration range.

In contrast, *S. tuberosum* exhibited a much steeper decline in viability. At lower concentrations (1 × 10⁻^11^–1 × 10⁻^7^ mg/mL), viability remained above 90%, but a sharp drop occurred at 1 × 10⁻⁵ mg/mL, reducing viability to 85%. This steep decline continued, with 60% viability at 1 × 10⁻^4^ mg/mL and 10% at 1 × 10⁻^3^ mg/mL, leading to 0% viability at 1 × 10⁻^2^ mg/mL. The IC_50_ value for *S. tuberosum* was significantly lower, at 0.070 mg/mL (95% CI: 0.061–0.079 mg/mL), with a much steeper slope of −41.83. These results indicate a concentration-dependent cytotoxic effect for both extracts, with *S. tuberosum* showing greater potency at lower concentrations compared to *Z. mays* (Figure 8).

The comparison of IC_50_ values highlights the greater potency of *S. tuberosum*, which achieved similar cytotoxic effects at significantly lower concentrations than *Z. mays*. The IC_50_ for *S. tuberosum* (0.070 mg/mL) was markedly lower than that of *Z. mays* (0.275 mg/mL). The increased potency observed in *S. tuberosum* L. may be attributable to its higher total polyphenol content (TPC) and elevated anthocyanin levels (Table 1). The greater cytotoxicity exhibited by *S. tuberosum* in HeLa cells, as indicated by its lower IC_50_ value, is likely a direct result of its enriched polyphenol and anthocyanin concentrations. These compounds are well-established for their cytotoxic properties across various cancer cell lines, including HeLa cells [60,61,62,63], and their presence reinforces the potential of *S. tuberosum* as a potent source of bioactive molecules for cancer therapy.

Further differences were apparent in the slopes of the dose–response curves, with *Z. mays* exhibiting a slope of −7.32 and *S. tuberosum* showing a much steeper slope of −41.83. The steeper slope for *S. tuberosum* reflects a more rapid and pronounced reduction in cell viability over a narrow concentration range, suggesting a highly efficacious cytotoxic mechanism. This sharper response could be attributable to the higher total polyphenol content (TPC) and anthocyanin levels in *S. tuberosum*, which are likely driving more potent bioactive effects (Table 1). The steep slope associated with *S. tuberosum* suggests a sharp therapeutic window, where small variations in concentration around the IC_50_ could lead to substantial biological effects.

The differences in potency and slope between the two extracts may be attributable to their distinct anthocyanin compositions. *Z. mays* is known to be rich in cyanidin-based anthocyanins, which have shown moderate cytotoxicity in various cancer models. In contrast, *S. tuberosum* may contain delphinidin or petunidin derivatives, which could account for its stronger cytotoxic effects [61]. Differences in anthocyanin structure, including glycosylation and acylation, play a critical role in determining their bioactivity [64].

Additionally, the observed differences in cytotoxicity may stem from factors such as solubility, stability, and bioavailability of the anthocyanins in each extract. Anthocyanins are sensitive to environmental factors such as pH and temperature [65,66], which can affect their stability and efficacy. It is possible that *S. tuberosum* anthocyanins are more stable or more readily absorbed by cells, leading to their more potent cytotoxic effects. Conversely, *Z. mays* anthocyanins may degrade more rapidly, resulting in a slower and less pronounced reduction in cell viability.

To investigate whether apoptosis was the primary mechanism behind the cytotoxic effects observed with both microencapsulated anthocyanins, the apoptotic index (AI) was analyzed by measuring the expression of pro-apoptotic markers Bax and Caspase-8 relative to the anti-apoptotic marker Bcl-2. The AI provided a measure of the balance between cell death promotion and inhibition (Appendix A). For *Z. mays* L., the control group showed an AI of 2.32 (±1.02), while treatment with the microencapsulated anthocyanins resulted in a slightly higher AI of 2.47 (±0.95). Although this increase indicates a modest rise in pro-apoptotic activity, the marginal difference, and overlapping standard deviations suggest that apoptosis is not the primary mechanism underlying the cytotoxic effects of *Z. mays* L. anthocyanins. While apoptosis likely contributes, these data imply that *Z. mays* L.’s cytotoxicity results from the interplay of multiple pathways.

Treatment with *S. tuberosum* L. anthocyanins increased the AI to 2.60 (±0.90), suggesting a stronger apoptotic response compared to *Z. mays* L. Nonetheless, the cytotoxic effects observed in previous studies likely involve additional mechanisms beyond apoptosis. The relatively high standard deviations in AI, coupled with the pronounced reduction in cell viability (Appendix A), point to other processes—such as oxidative stress or mitochondrial dysfunction—as critical contributors to *S. tuberosum* L. anti-tumor effects.

These findings indicate that while both *Z. mays* L. and *S. tuberosum* L. anthocyanin extracts induce apoptosis in HeLa cells, apoptosis is unlikely to be the sole driver of their cytotoxic effects. Anthocyanins exhibit a range of biological activities that contribute to their antitumoral effects, including inhibition of cell proliferation, suppression of angiogenesis, and modulation of signaling pathways. These multifaceted actions suggest that while apoptosis is a significant mechanism, it is not the major pathway through which anthocyanins exert their anticancer effects [67,68].

Further investigation into the molecular pathways affected by these anthocyanin extracts is warranted to elucidate their cytotoxic mechanisms fully. Exploring the interplay between apoptosis and other processes—such as autophagy, necrosis, or cell cycle disruption—will be crucial in optimizing anthocyanin-based therapeutic strategies for cancer treatment.

## 4. Conclusions

This study provides comprehensive insights into the biological potential of microencapsulated anthocyanins derived from *S. tuberosum* and *Z. mays*, emphasizing their relevance as sources of bioactive compounds with therapeutic and functional food applications. The superior antioxidant activity of *S. tuberosum*, coupled with its strong cytotoxic effects against HepG2 and THJ29T cancer cell lines, highlights its promise as a natural therapeutic agent for oxidative stress-related diseases and cancer. On the other hand, *Z. mays* demonstrated notable antimicrobial activity against clinically relevant multidrug-resistant bacterial strains and biofilm-forming pathogens, addressing critical global health challenges such as antimicrobial resistance. The use of microencapsulation significantly improved the stability and bioavailability of anthocyanins, as evidenced by FTIR and SEM analyses, providing a robust platform for enhancing their functional properties. These findings reinforce the value of ancestral Andean edible plants as untapped reservoirs of bioactive molecules, paving the way for their integration into modern nutraceuticals and pharmaceutical formulations. Future studies should prioritize in vivo validations, detailed mechanistic investigations, and the development of optimized formulations to maximize their therapeutic potential and applicability in diverse fields, including healthcare and food sciences.

## Figures and Tables

**Figure 1 nutrients-16-04078-f001:**
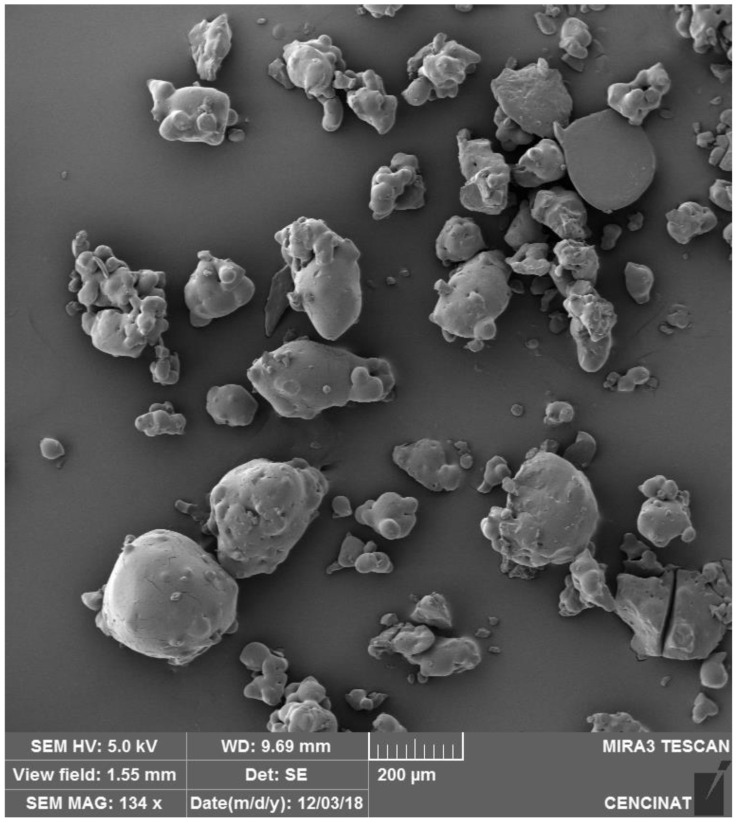
SEM image of *S. tuberosum* L. microencapsulated particles prepared by spray drying, captured with a TESCAN MIRA 3 SEM at 134× magnification and a 200 µm scale bar.

**Figure 2 nutrients-16-04078-f002:**
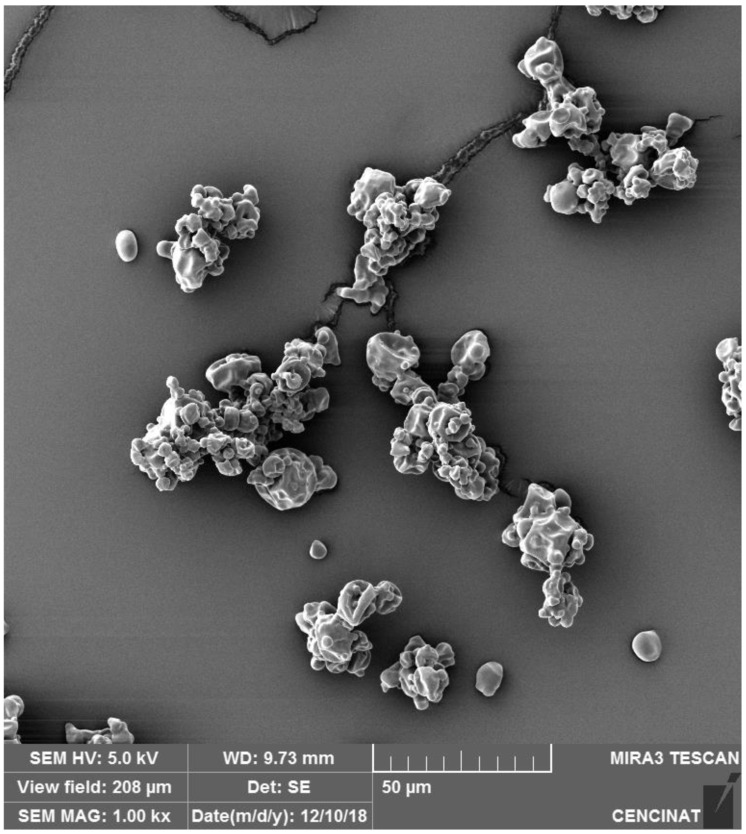
SEM image of *Z. mays* L. microencapsulated particles prepared by spray drying, captured with a TESCAN MIRA 3 SEM at 1000× magnification and a 50 µm scale bar.

**Figure 3 nutrients-16-04078-f003:**
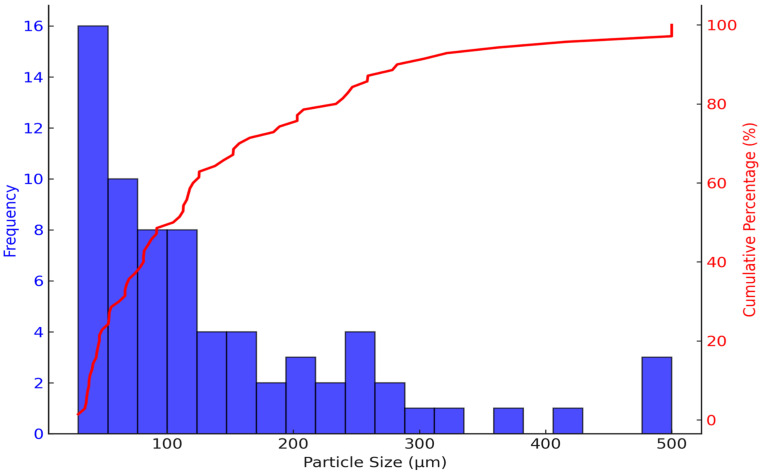
Particle size distribution analysis of *S. tuberosum* L. microencapsulated spheres. Particle size distribution and cumulative distribution curve of *S. tuberosum* L microencapsulated spheres, measured using FIJI software (version 2.9.0).

**Figure 4 nutrients-16-04078-f004:**
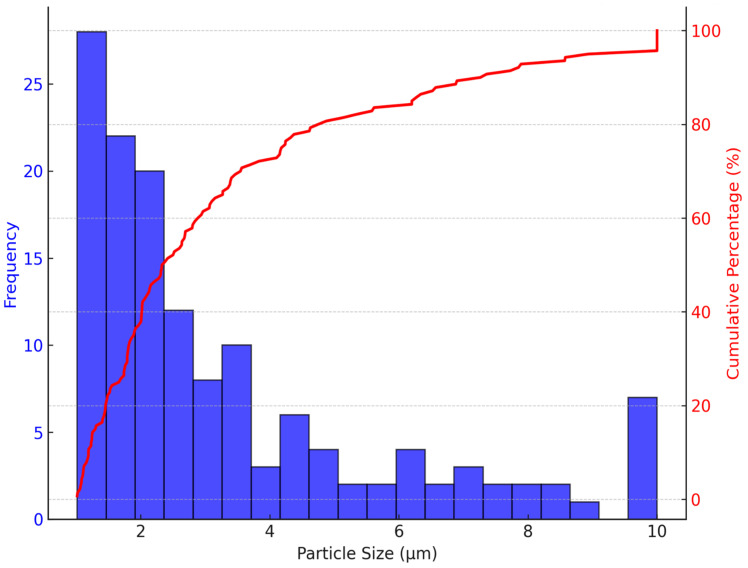
Particle size distribution analysis of *Z. mays* L. microencapsulated spheres. Particle size distribution and cumulative distribution curve of *Z. mays* microencapsulated spheres were measured using FIJI software (version 2.9.0).

**Figure 5 nutrients-16-04078-f005:**
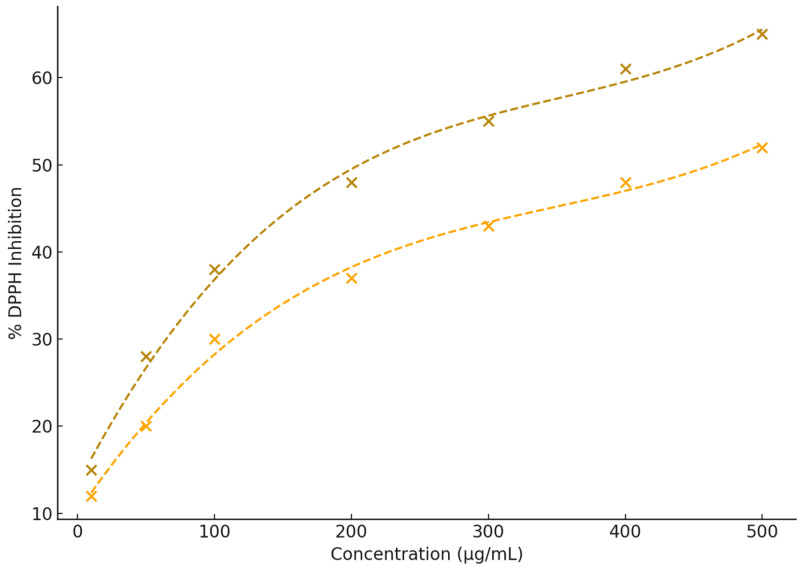
Antioxidant activity (% DPPH inhibition) of microencapsulated anthocyanins from *S. tuberosum* L. and *Z. mays* L. The plot illustrates the relationship between anthocyanin concentration and % DPPH inhibition. Data points represent the mean % DPPH inhibition, with *S. tuberosum* depicted as dark goldenrod cross and *Z. mays* as orange cross. Cubic polynomial fits (dashed lines).

**Figure 6 nutrients-16-04078-f006:**
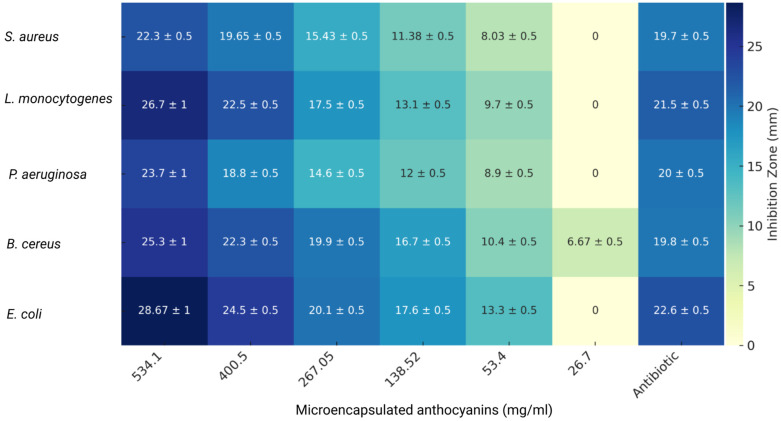
Antimicrobial activity of microencapsulated *S. tuberosum* anthocyanin extract against different bacterial strains. Heatmap showing the inhibition zones (in mm) with standard deviations for *S. aureus*, *L. monocytogenes*, *P. aeruginosa*, *B. cereus*, and *E. coli* at various concentrations (534.1 to 26.7 mg/mL) of the microencapsulated *S. tuberosum* extract. Antibiotic (gentamicin) was used at the standard concentration.

**Figure 7 nutrients-16-04078-f007:**
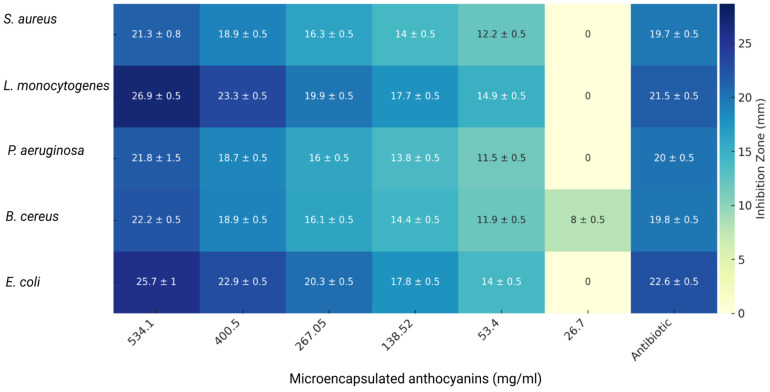
Antimicrobial activity of microencapsulated *Z. mays* anthocyanin extracts against different bacterial strains. Heatmap showing the inhibition zones (in mm) with standard deviations for *S. aureus*, *L. monocytogenes*, *P. aeruginosa*, *B. cereus*, and *E. coli* at various concentrations (534.1 to 26.7 mg/mL) of the microencapsulated *S. tuberosum* extract. Antibiotic (gentamicin) was used at the standard concentration.

**Figure 8 nutrients-16-04078-f008:**
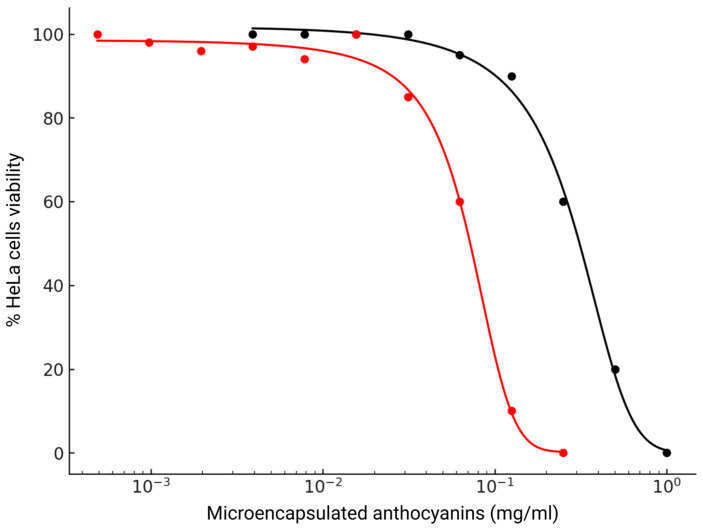
Dose–response curves of the cytotoxic effects of *Z. mays* and *S. tuberosum* L. The *X*-axis (log scale) represents the concentration of anthocyanin extract in mg/mL, while the *Y*-axis shows the percentage of viable cells. Data points for *Z. mays* are depicted as black circles, with the fitted sigmoid curve shown as a solid black line. Red circles denote data points for *S. tuberosum*, with the corresponding fitted curve represented by a red line.

**Table 1 nutrients-16-04078-t001:** Polyphenol and anthocyanin content of *S. tuberosum* L. and *Z. mays* L.

Parameter	*S. tuberosum*	*Z. mays*
TPC (mg of gallic acid equivalents (GAE) per kg of fresh weight (fw))	554 * ± 25.16	494 * ± 24.03
Anthocyanin content (mg C3G/100 g)	71.40	63.5

TPC: Total polyphenols content. * The asterisk (*) indicates that the difference between samples is statistically significant (*p* < 0.001).

**Table 2 nutrients-16-04078-t002:** Summary of FTIR Comparative Visual Analysis of *Z. mays* L. Comparative visual analysis of FTIR spectra of non-microencapsulated and microencapsulated anthocyanins from *Z. mays* L.

Wavenumber (cm⁻^1^)	Functional Group	Non-Microencapsulated (Qualitative Intensity)	Microencapsulated (Qualitative Intensity)
3200–3600	O-H stretching (hydroxyl)	High	Medium
2800–3000	C-H stretching (alkanes)	Medium	Low
1700–1750	C=O stretching (carbonyl)	High	Low
1500–1600	C=C stretching (aromatic rings)	Medium	Low
1000–1300	C-O, C-O-C stretching (ethers, glycosidic bonds)	Medium	Low
1027	C-O-C stretching (polysaccharides, matrix interactions)	-	Medium
840–760	New peaks (maltodextrin matrix vibrations)	-	Medium

**Table 3 nutrients-16-04078-t003:** Summary of FTIR Comparative Visual Analysis of *S. tuberosum* L. Comparative visual analysis of FTIR spectra of non-microencapsulated and microencapsulated anthocyanins from *S. tuberosum* L.

Wavenumber (cm⁻^1^)	Functional Group	Non-Microencapsulated (Qualitative Intensity)	Microencapsulated (Qualitative Intensity)
3200–3600	O-H stretching (hydroxyl)	High	Medium
2800–3000	C-H stretching (alkanes)	Low	Medium
1700–1750	C=O stretching (carbonyl)	High	Low
1500–1600	C=C stretching (aromatic rings)	Medium	Low
1000–1300	C-O, C-O-C stretching (ethers, glycosidic bonds)	Medium	Low
1027	C-O-C stretching (polysaccharides, matrix interactions)	-	Medium
840–760	New peaks (maltodextrin matrix vibrations)	-	Medium

**Table 4 nutrients-16-04078-t004:** MIC of Microencapsulated anthocyanin extracts.

Bacterial Strain	MIC (mg/mL)	
	*S. tuberosum*	*Z. mays*
*S. aureus*	53.4	53.4
*L. monocytogenes*	53.4	53.4
*P. aeruginosa*	53.4	53.4
*B. cereus*	26.7	26.7
*E. coli*	53.4	53.4

## Data Availability

The original contributions presented in the study are included in the article/Appendix A, further inquiries can be directed to the corresponding author.

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
