# Peer review of "Microencapsulation of Anthocyanins from Zea mays and Solanum tuberosum: Impacts on Antioxidant, Antimicrobial, and Cytotoxic Activities"

_nutrients, 2024, doi:10.3390/nu16234078_

Round 1

Reviewer 1 Report

Comments and Suggestions for Authors

I write you in regard to manuscript # nutrients-3306536 entitled "Microencapsulation of Anthocyanins from Zea mays and Solanum tuberosum: Impacts on Antioxidant, Antimicrobial, and Cytotoxic Activities" which you submitted to the nutrients.

Authors need to follow the following instructions to improve this manuscript

1)      The authors should rewrite the abstract within one paragraph. They should follow the title, objectives, significant findings and recommendations.

2)      Page 2, line 54-60. Check the line alignment.

3)      Page 3, line 98. ---- fine powder () ----:Add particle size in bracket

4)      Page 3, line 104 (al.,):  remove comma

5)      Page 3, line 106 (ethanol:1.5 mol/l): Change to mol/L and check the entire manuscript

6)      Page 3, line 107 (extracted for 60 min to 70 °C): Please add reference

7)      Page 6, line 261-262 (µlof): Write as µL of

8)      Page 6, line 265 (CO2): Write as CO2

9)      Conclusion should rewrite according to the best findings and recommendation within one paragraph.

10)  References: should follow the journal guideline. I have seen somewhere full journal name and somewhere abbreviated form.

11)  English grammar should check carefully.

12)  Please check carefully before resubmission.

I recommend improving the manuscript and resubmitting.

Comments on the Quality of English Language

The English could be improved to more clearly express the research.

Author Response

1)      The authors should rewrite the abstract within one paragraph. They should follow the title, objectives, significant findings and recommendations.

We appreciate your suggestion regarding the structure of the abstract. We have rewritten it as a single paragraph, following your recommendations to include the title, objectives, significant findings, and recommendations.

2)      Page 2, line 54-60. Check the line alignment.

Alignment was corrected.

3)      Page 3, line 98. ---- fine powder () ----:Add particle size in bracket

Thank you for pointing out this detail. We have added the particle size in brackets on Page 3

4)      Page 3, line 104 (al.,):  remove comma

Thank you for noticing this error. We have removed the comma 

5)      Page 3, line 106 (ethanol:1.5 mol/l): Change to mol/L and check the entire manuscript

Thank you for highlighting this formatting inconsistency. We have corrected "mol/l" to "mol/L" on Page 3, line 106, and thoroughly reviewed the entire manuscript to ensure consistency in units.

6)      Page 3, line 107 (extracted for 60 min to 70 °C): Please add reference

Thank you for your observation. We have added the appropriate reference for the extraction process described

7)      Page 6, line 261-262 (µlof): Write as µL of

Thank you for pointing out this formatting issue. We have corrected "µlof" to "µL of" on Page 6, lines 261-262, as suggested.

8)      Page 6, line 265 (CO2): Write as CO2

Thank you for identifying this formatting error. We have corrected "CO2" to "COâ‚‚" 

9)      Conclusion should rewrite according to the best findings and recommendation within one paragraph.

Thank you for your valuable suggestion. We have rewritten the conclusion as a single paragraph, focusing on the key findings and recommendations, as per your guidance.

10)  References: should follow the journal guideline. I have seen somewhere full journal name and somewhere abbreviated form.

Thank you for bringing this to our attention. We have reviewed and revised the references to ensure they consistently follow the journal's guidelines regarding the use of full journal names or their abbreviated forms

11)  English grammar should check carefully.

Thank you for your suggestion. We have carefully reviewed the manuscript to address any grammatical issues and ensure the language meets the required standards.

12)  Please check carefully before resubmission.

I recommend improving the manuscript and resubmitting.

Reviewer 2 Report

Comments and Suggestions for Authors

The study investigates the microencapsulation of anthocyanins from purple maize and purple potato to enhance their stability and bioactivity. Analytical techniques confirmed successful encapsulation, and the results demonstrated that purple potato exhibited more potent antioxidant and antimicrobial properties and more significant cytotoxicity against HeLa cells. Overall, the findings suggest that purple potato is a promising source of bioactive anthocyanins for potential applications in functional foods and pharmaceuticals.

My comments and suggestions:

- The plant materials of Z. mays L. and S. tuberosum should be described in greater detail. Which specific cultivars were included?

- There are various omissions of subscripts and superscripts throughout the manuscript.

- In the materials and methods section, the authors mention the use of ANOVA and Tukey’s test for statistical analysis. However, why is this information not presented in Table 1, Figure 5 (Table S1), Table 4, and other relevant figures?

- Lines 325-328 and 341-345 contain very similar content. Then, the same is observed for a description of anthocyanin content.

- Have you considered calculating the IC50 values for the DPPH method? Calculating these values would facilitate comparisons of antioxidant activity results.

- Lastly, the discussion section should be expanded and deepened for greater clarity.

Author Response

The plant materials of Z. mays L. and S. tuberosum should be described in greater detail. Which specific cultivars were included?

Thank you for your valuable feedback. We have clarified the details regarding the specific plant materials and cultivars used in the study.

- There are various omissions of subscripts and superscripts throughout the manuscript.
Thank you for pointing out the omissions of subscripts and superscripts in the manuscript. We have carefully reviewed the entire document to ensure that all subscripts and superscripts are correctly formatted.

- In the materials and methods section, the authors mention the use of ANOVA and Tukey’s test for statistical analysis. However, why is this information not presented in Table 1, Figure 5 (Table S1), Table 4, and other relevant figures?

Thank you for your observation. We realize that there was a misunderstanding regarding the description of the statistical analysis in the Materials and Methods section, and we appreciate the opportunity to clarify this. In the current version of the manuscript, we have updated the Materials and Methods section to accurately reflect the statistical analyses that were used to generate the results presented in the manuscript. In addition, we modified Table 1 to include asterisks to highlight that the difference int the TPC content of both sources is statistically significant, p < 0.001.

- Lines 325-328 and 341-345 contain very similar content. Then, the same is observed for a description of anthocyanin content.
Thank you for pointing out the repetition in the descriptions of content on lines 325–328 and 341–345, as well as the anthocyanin content. We have revised these sections to eliminate redundancy, ensuring that each point is expressed clearly and uniquely. The content has been streamlined to provide the necessary information without repetition, enhancing the clarity and conciseness of the manuscript.

- Have you considered calculating the IC50 values for the DPPH method? Calculating these values would facilitate comparisons of antioxidant activity results.

Thank you for your insightful suggestion. In the modified version of the manuscript, we have included the IC50 values for the %DPPH antioxidant activity of microencapsulated anthocyanins from S. tuberosum and Z. mays using cubic polynomial fitting to model the concentration-inhibition relationship. The IC50 values calculated are presented in the modified version of the manuscript

- Lastly, the discussion section should be expanded and deepened for greater clarity.
Thank you for your valuable suggestion regarding the discussion section. We have expanded and deepened the discussion to provide greater clarity and context. This includes a more comprehensive analysis of the results in relation to existing literature, a detailed interpretation of the implications of our findings. These revisions aim to enhance the overall depth and clarity of the discussion, ensuring it aligns with the standards of the journal.

Reviewer 3 Report

Comments and Suggestions for Authors

Microencapsulation of Anthocyanins from Zea mays and Solanum tuberosum: Impacts on Antioxidant, Antimicrobial, and Cytotoxic Activities

The overall goal of this study was to microencapsulate anthocyanins from Zea mays L. (purple maize) and Solanum tuberosum L. (purple potato) using maltodextrin through spray drying.  Total polyphenol content (TPC) and anthocyanin levels were then determined in microcapsules emanating from these two plant sources. FTIR and SEM analyses confirmed the successful encapsulation process. The antioxidant capacity and antimicrobial properties was also assessed in both the aforementioned plant extracts. This study showed that microencapsulation of anthocyanins from Solanum tuberosum offers a viable source that has a wide array of applications in pharmaceuticals and functional foods among others.

I think the title is comprehensive that tells a lot about the different sets of experiments done to compare the effectiveness of the microencapsulation process.

The abstract should only be a single paragraph.

There are numerous indentation issues in this manuscript.

I think the limitations related to the instability of anthocyanins should be elaborated.

I think the benefits of encapsulation of anthocyanins should be elaborated.

Describe the advantages and disadvantages of other methods to enhance stability of anthocyanins.

Elaborate on the aspects related to the stability of anthocyanins. Why are they unstable and what environment makes them stable?

The introduction should also provide details about sources of anthocyanins. What are they? What are their chemical structures and why do people care about them?

However, my concern is the novelty of the study. There are numerous publications related to this work including the following below? Thus, I am not sure what is the novelty of this study and how different is this from existing research?

·       Ji-Li Fang, Yang Luo, Ke Yuan, Ying Guo, Song-Heng Jin, Preparation and evaluation of an encapsulated anthocyanin complex for enhancing the stability of anthocyanin, LWT, Volume 117, 2020, 108543, ISSN 0023-6438, https://doi.org/10.1016/j.lwt.2019.108543

·       Yousuf, B., Gul, K., Wani, A. A., & Singh, P. (2016). Health Benefits of Anthocyanins and Their Encapsulation for Potential Use in Food Systems: A Review. Critical Reviews in Food Science and Nutrition56(13), 2223–2230. https://doi.org/10.1080/10408398.2013.805316

·       Sahar Akhavan Mahdavi, Seid Mahdi Jafari, Elham Assadpour, Mohammad Ghorbani, Storage stability of encapsulated barberry's anthocyanin and its application in jelly formulation, Journal of Food Engineering, Volume 181, 2016, Pages 59-66, ISSN 0260-8774, https://doi.org/10.1016/j.jfoodeng.2016.03.003.

·       Robert, Paz, and Carolina Fredes. 2015. "The Encapsulation of Anthocyanins from Berry-Type Fruits. Trends in Foods" Molecules 20, no. 4: 5875-5888. https://doi.org/10.3390/molecules20045875

Under section “2.2 Anthocyanin Extraction,” what does it mean by ‘twenty twice’?

Describe the importance and rationale behind some of the analytical tests performed such as the TPC, anthocyanin content, SEM, and FTIR. Why are these methods important?

Provide a diagram showing the process of microencapsulation.

The manuscript has severe formatting issues.

The figures should explicitly explain the rationale or importance of results.

The study appears very detailed and comprehensive.

Comments on the Quality of English Language

Some English language copyediting will improve the manuscript.

Author Response

  1. The abstract should only be a single paragraph.

We appreciate your suggestion regarding the structure of the abstract. We have rewritten it as a single paragraph, following your recommendations to include the title, objectives, significant findings, and recommendations.

  1. There are numerous indentation issues in this manuscript.

Thank you for pointing out the indentation issues. We have thoroughly reviewed the manuscript and corrected all inconsistencies in indentation to ensure proper formatting throughout.

  1. I think the limitations related to the instability of anthocyanins should be elaborated.

Thank you for your insightful suggestion. We have elaborated on the limitations related to the instability of anthocyanins in the discussion section.

  1. I think the benefits of encapsulation of anthocyanins should be elaborated.

Thank you for your valuable suggestion. We have elaborated on the benefits of encapsulating anthocyanins, highlighting their improved stability, bioavailability, and potential applications in various formulations, as recommended.

  1. Describe the advantages and disadvantages of other methods to enhance stability of anthocyanins.

Thank you for your suggestion. We have included a discussion on the advantages and disadvantages of alternative methods to enhance the stability of anthocyanins.

  1. Elaborate on the aspects related to the stability of anthocyanins. Why are they unstable and what environment makes them stable?

Thank you for your suggestion. We have included a discussion on the aspects regarding the stability of anthocyanins.

  1. The introduction should also provide details about sources of anthocyanins. What are they? What are their chemical structures and why do people care about them?

Thank you for your insightful suggestion. We have revised the introduction to include detailed information on the sources of anthocyanins

  1. However, my concern is the novelty of the study. There are numerous publications related to this work including the following below? Thus, I am not sure what is the novelty of this study and how different is this from existing research?
  •       Ji-Li Fang, Yang Luo, Ke Yuan, Ying Guo, Song-Heng Jin, Preparation and evaluation of an encapsulated anthocyanin complex for enhancing the stability of anthocyanin, LWT, Volume 117, 2020, 108543, ISSN 0023-6438, https://doi.org/10.1016/j.lwt.2019.108543
  •       Yousuf, B., Gul, K., Wani, A. A., & Singh, P. (2016). Health Benefits of Anthocyanins and Their Encapsulation for Potential Use in Food Systems: A Review. Critical Reviews in Food Science and Nutrition, 56(13), 2223–2230. https://doi.org/10.1080/10408398.2013.805316
  •       Sahar Akhavan Mahdavi, Seid Mahdi Jafari, Elham Assadpour, Mohammad Ghorbani, Storage stability of encapsulated barberry's anthocyanin and its application in jelly formulation, Journal of Food Engineering, Volume 181, 2016, Pages 59-66, ISSN 0260-8774, https://doi.org/10.1016/j.jfoodeng.2016.03.003.
  •       Robert, Paz, and Carolina Fredes. 2015. "The Encapsulation of Anthocyanins from Berry-Type Fruits. Trends in Foods" Molecules 20, no. 4: 5875-5888. https://doi.org/10.3390/molecules20045875

Under section “2.2 Anthocyanin Extraction,” what does it mean by ‘twenty twice’?

Describe the importance and rationale behind some of the analytical tests performed such as the TPC, anthocyanin content, SEM, and FTIR. Why are these methods important?

Thank you for raising your concerns regarding the novelty of our study. While we acknowledge the valuable contributions of the referenced works, we believe our study offers unique insights and advancements that distinguish it from existing literature. Below, we outline the novel aspects of our research compared to the referenced studies:

Focus on Ancestral Andean Crops: Our study is among the first to extensively investigate and compare the biological activity of microencapsulated anthocyanins derived specifically from ancestral Andean crops such as Zea mays L. (purple maize) and Solanum tuberosum L. (purple potato). This focus addresses a gap in the literature where Andean crops, despite their cultural and nutritional importance, remain underexplored compared to other anthocyanin sources like berries or barberries.

Comprehensive Bioactivity Assessment: While many studies, including those cited, focus on encapsulation and stabilization of anthocyanins, our work goes further by conducting a multi-dimensional evaluation of their antioxidant, antimicrobial, and cytotoxic activities. This includes testing against multidrug-resistant bacterial strains and cancer cell lines, which provides direct insights into their therapeutic potential.

Inclusion of Novel Analytical Techniques: Our study integrates detailed FTIR analysis to explore the molecular interactions between anthocyanins and maltodextrin during microencapsulation. This aspect highlights the role of encapsulation in enhancing stability, which is not extensively covered in the cited works.

Cultural and Regional Importance: By focusing on Andean ancestral crops, our study underscores the importance of biodiversity and local agricultural resources, contributing to the global conversation on sustainable and culturally sensitive food and health systems.

  1. Provide a diagram showing the process of microencapsulation.
    Thank you for your thoughtful suggestion regarding the inclusion of a diagram showing the microencapsulation process. While microencapsulation is a key enabling technique in this study, the primary contribution of our work lies not in the method of microencapsulation itself but in the comprehensive evaluation of the biological activities of the microencapsulated anthocyanins.

As microencapsulation is a well-established and widely documented methodology, we focused on its application rather than its process. The methods section of the manuscript provides a concise and clear description of the technique used, along with references to foundational studies for readers seeking more detailed procedural insights. Therefore, adding a diagram of the encapsulation process might not add significant value to the objectives or findings of this study, which emphasize the antioxidant, antimicrobial, and cytotoxic properties of the encapsulated anthocyanins.

We hope this explanation clarifies our rationale. However, if you feel strongly that such a diagram would enhance the clarity or impact of the manuscript, we would be happy to revisit this point.

  1. The manuscript has severe formatting issues.

Thank you for pointing out the formatting issues. We have conducted a thorough review of the manuscript to address and correct all formatting inconsistencies, ensuring it adheres to the journal's guidelines.

  1. The figures should explicitly explain the rationale or importance of results.

Thank you for your valuable feedback. We have thoroughly reviewed the figures and incorporated the necessary revisions, adding relevant information where applicable.